# Dust Particle Counter for Powder Bed Fusion Process

**DOI:** 10.3390/s22197614

**Published:** 2022-10-08

**Authors:** Stanisław Karcz, Grzegorz Skrabalak, Andrzej Brudnik, Grzegorz Gajoch

**Affiliations:** 1Łukasiewicz Research Network—Krakow Institute of Technology, Zakopiańska 73 Street, 30-418 Cracow, Poland; 2Institute of Electronics, AGH University of Science and Technology, Al. Mickiewicza 30, 30-059 Cracow, Poland

**Keywords:** air particle counter, laser sensor, powder bed fusion, additive manufacturing

## Abstract

The paper presents a novel dust detector based on an innovative laser system that can be successfully used in applications where continuous dust monitoring is necessary. The measurements obtained with FeNi18Co9Mo5 (maraging MS1 steel) particles are compared with the particle fall times calculated using the Navier–Stokes equation. The measurement powder was subjected to sieve analysis and laser system detection. Based on the results obtained, a formula was developed to determine the dust concentration depending on the number and size of particles. With filtration applied, the detector measurement range was from 16 to 100 µm. The developed solution can be the basis for the development of a dedicated sensor for powder bed fusion processes.

## 1. Introduction

The recent development and widespread usage of additive manufacturing techniques offer a way to address the need to provide safe working conditions for persons involved in material processing. For example, for commonly used fused deposition modeling (FDM) and digital light processing (DLP) technologies, it is necessary to pay more attention to matters related to the volatile organic compounds that are produced during the melting of polymeric materials [1,2]. In the case of powder bed fusion manufacturing processes (i.e., selective laser sintering/melting—SLS/SLM, electron beam melting—EBM, binder jetting—BJ), the main hazardous aspect concerns potential contact of the system operator with powders of sizes below 100 µm and dust measured in the micro and nanometer range. The permitted levels of volatile organic compounds (VOC) and dust values at workplaces are regulated by the laws (H&S regulations) of each country and, depending on the type of hazard, define safe exposure concentrations and durations [3]. The important problem with the above-mentioned hazardous substances and compounds concerns the systems for measurements of dust and/or volatile organic compound concentrations, which are not continuously monitored [4].

The results of the dust measurement systems market analysis showed that there are no commercially available systems dedicated for dust and powder measurement in the case of powder bed additive manufacturing processes. There is only one laser sensor module available, which costs around USD 250. According to its specifications, the sensor allows for the measurement of air-suspended particles up to 100 µm [5], however, the manufacturer does not specify the maximum density of the particles, which makes it difficult to apply this module for the powders used in additive manufacturing processes.

Dust measurements at workplaces are usually carried out once to determine the workplace risk or the maximum time of exposure to a given substance. Unfortunately, there are no continuous dust monitoring systems that are designed for use during the additive materials stage in manufacturing processes. There are personal protection devices available that can operate continuously, but as their price is very high, they are rarely used [6,7]. On the other hand, on the market, there are numerous sensors and solutions for smog detection available (particles in the range of PM2.5 and PM10). Due to the variety of materials used in the powder-based manufacturing processes and their high density, different sensors and measurement solutions other than those used in the case of PM2.5 and PM10 must be applied [8]. 

Measurements of dustiness in industrial environments are very important as the exposure of workers to hazardous factors resulting from materials used and the harsh environment may cause occupational diseases [9]. Contact with metallic powders and improper protective gear may cause absorption of substances through the skin or by inhalation of particular powder fractions [10]. Considering the fact that not only in the processes of additive manufacturing are the employees exposed to dust containing metallic compounds, but dust surveys are also carried out at workstations where metals and metallic powders are processed. On the basis of the results of the conducted research, it is intended to develop standards for risk assessments at employee workstations [9,11].

Dust concentration in the area of the equipment used for powder bed fusion processes may vary depending on the technology used. During the preparation of the material for the process, it is necessary to sieve the powder or mix it, which can result in a critical increase in dust. In the case of powder bed fusion equipment and the use of different materials for additive manufacturing, a thorough screening of the powder is necessary to get rid of agglomerates that can form [12,13]. Depending on the design and type of the machine and the cleaning and screening process, the exposure of the operator to hazardous substances can occur for up to several hours.

During these operations, it is necessary to use protective gear to limit the contact of the powder through the skin and the amount of inhaled particles by using masks with H13 or H14 class filters. The use of dust masks with fine filters results in high breathing resistance and discomfort during physical effort. In industrial use, it is technically possible to transport the powder inside the machine with compressed air and to limit dust exposure with a single material and repeatable production. The last critical stage of the process is the finishing, where not fully sintered powder grains are removed. In the case of the SLS and SLM processes, this is done by sandblasting or shot peening. The use of compressed air can cause even heavy dusts to rise, which can be hazardous to the operator.

A continuous and economically available dust measurement system would allow for the measurement of the presence and level of hazardous substances in workshops with powder-based additive manufacturing systems, as well as help to define safe and unsafe zones where protective masks and clothing must be used.

Based on the information available in the literature, in order to develop a device for dust monitoring, the properties of the powder particles must be taken into account, such as shape, material density, and diameter of the expected particles. For proper functioning of the dust sensor, it is necessary to know the shape, size, and density of the material in µg/m^3^. If no powder parameters are given, it is possible to provide the result in the number of particles per given volume of air, which can also be an indicator of air quality.

The measurement of dust with diameters between 10 and 100 µm in the work environment is very important, as there is currently no obligation to use continuous dust measurement equipment at workstations for powder bed technologies and other additive manufacturing technologies. Currently, there are no low-cost solutions in this area. Therefore, the article presents an innovative detector ‘AS one’, which is effective in this type of research.

The developed measurement system includes commercial and proprietary detectors. The system is intended to measure dust with diameters ranging from 15 to 100 µm. Commercial detectors use the effect of laser diffraction on single particles or groups of particles. The measurement algorithm of factory-made modules is unknown. The presented ‘AS one’ detector uses the phenomenon of laser diffraction, taking into account the diameter of the particles and the density of the sucked material.

The article compares measurement results obtained from the ‘AS one’ detector and the SDS198 modules. The measurement system and results are presented later in this paper.

### 1.1. Particle Characteristics

Most metallic powders are produced in the same way, by the process of gas or water atomisation [14]. Not all particles have a spheroidal shape [15], additionally, in the case of SLS/SLM, EBM or DED (Direct Energy Deposition) processes, some of the powder particles lose their spheroidal shape after heating [16]. Due to its properties, one of the most commonly used materials in the tooling industry, which is processed using additive manufacturing methods—mainly SLS/SLM—is MS1 steel. It is characterized by high hardness [17,18] (over 50 HRC after heat treatment). MS1 steel is used for the production of injection moulds with conformal cooling channels, which allow for the shortening of mould cooling and injection cycle times [19,20,21]. The shape and grain size of the powder have a great influence on the sintering kinetics [22]. Despite the controlled pulverisation and atomisation process, the distribution in terms of sphericity and grain size is a normal distribution. There are processes of powder bed fusion (PBF) in which particle shape is not important. An example of such a powder process is binder jetting (BJ), where both the size and shape of the powder grains can almost be arbitrary [23]. The initial binding with a binder is followed by sintering in a furnace. Despite the two-step manufacturing system, the BJ process can be more efficient than the SLM or SLS process.

Additive manufacturing in PBF technologies incorporating the effect of high energy heating usually takes place in an inert gas atmosphere and in closed, sealed chambers, which significantly reduces the spread of dust. The highest dust concentrations are found during postprocessing [24], where it is required to remove the element from the powder volume and clean off the loose powder granules.

PBF technologies are becoming increasingly popular due to the development of additive technologies [25] and the increasing availability of materials in powder form [26]. The PBF technologies allow for almost any shape to be obtained with minimal or no supports (in the case of polymers) [27].

Particle sizes in PBF processes range from 20 to 65 µm [15]. Unfortunately, some materials are very difficult to process by machining, e.g., silicon carbide or tungsten carbide [28,29]. Currently, the most popular PBF technologies are selective laser sintering/melting, in which polymer powders (SLS), usually obtained by pulverization, are rolled [30], and selective laser melting (SLM), where the most common material is MS1 maraging steel [31] (FeNi18Co9Mo5, US classification 18% Ni Maraging 300, European 1.2709 and German X3NiCoMoTi 18-9-5 [18]) and 316L stainless steel. The SLM process uses powders obtained by atomisation in gas. The atomised powder particles have a spheroidal shape [32].

### 1.2. Dust Detection Methods

The reference method used for the measurements is the gravimetric method which, unfortunately, is not useful for continuous dust measurement. Depending on the specifics of the workstation, the measurement may last from 15 min to 8 h [9,33]. Among dust and particle measurement devices, an important position is the use of laser sensors. They can be divided into two groups, analogue and digital. Analogue sensors scatter light on the dust cloud, while digital sensors count individual powder particles [33]. In practical applications for the illumination of particles, infrared light or visible light in the red light range is used due to the cost of the illuminator and the sensitivity of the photodiode acting as the detector. A very important aspect is also the angle between the detector and the light source [34].

The physical principle behind the particle size-dependent function of the sensor is the theory of Mie or Fraunhofer [35,36]. Both physical theories assume that the scattering of laser light on perfectly circular particles is due to unknown complex factors for the Mie equation and the simpler form of the Fraunhofer equation for larger particles. Unfortunately, automatic measurements performed with sensors based on optical measurements are not the same as the results obtained from measurements using the gravimetric method [33]. 

## 2. Materials and Methods

### 2.1. Measuring Stand

For sensor calibration and verification, a dedicated measurement stand/chamber was developed. The chamber allowed for measurements to be performed with or without forced air circulation, which was provided by fans placed inside the chamber. During experiments, each sensor sent data from a 180 s measurement cycle to the cloud, where the data were further processed (i.e., calculated the arithmetic mean of the measurements from three minutes). Additionally, pulses from the ‘AS one’ sensor were recorded on an oscilloscope in single trigger mode. Triggering was done manually every five minutes. After 8 h, 100 frames of 2.8 ms each of data were collected.

The operating principle of the test bench is shown in Figure 1.

The measurements were performed at 25 °C. The air humidity did not exceed 70% inside the test chamber.

The walls of the 1 m^3^ chamber (1 × 1 × 1 m) were made of glass (Figure 2). The chamber was equipped with a class 13 HEPA filter, which prevented powder from escaping the chamber. During the measurements, a slight overpressure was maintained in the chamber to avoid unwanted air “inhaling”. The measurement cables were routed through IP65-rated cable glands. 

Prior to performing experiments with the steel powder, the powder distribution characteristics were derived using sieve analysis. For sieve analysis, 100 g of MS1 was used. Sieve analysis was performed to determine the grain size distribution. From the collected sample, 20 g of powder was injected into the chamber with the use of a pneumatic cannon located in the chamber lid.

Four SDS198 detectors S1–S4 (Figure 2) operating in the continuous mode were used for the measurements. The measurement modules were located inside the chamber at heights of 20, 40, 60 and 80 cm above the chamber bottom. A stainless steel ladder was used to fix the sensors in the space. The ESP8266 modules (NodeMCU v3 [37]) transmitted data from the SDS198 [5] sensors directly to the cloud, where the results were visualised using Grafana software. The Grafana software allowed all data to be visualised directly and for operations such as plotting the average concentration over a period of time and determining measurement deviations between sensors to be performed.

The proprietary ‘AS one’ sensor was placed in the center of the chamber.

### 2.2. ‘AS One’ Dust Detector

The innovative sensor was made using FDM technology from black polylactide (PLA UltraPLA Noctuo Filaments). A cross-section of the sensor is shown in Figure 3.

The detector was based on a PIN photodiode (BPX 65). The angle between the illuminator and detector was approximately 28 degrees. The power of the laser diode of the illuminator was 5 mW, and the wavelength was 650 nm. The fan used to suck in the powder particles was 12 V and had a flow rate of 9.85 m^3^/h.

A PIN photodiode in a charge sense preamplifier (CSP) circuit [38,39,40] was used as a detector due to the low junction capacitance and short response time of the circuit. The PIN diode in a TO18 case was connected directly to the inverting input of the amplifier. OPA 350 operational amplifiers were used due to their low current, voltage noise and low input polarization current [41]. The total gain of the detector circuit was approximately 190 dBΩ. The bandwidth for the circuit was 16 kHz. The time constant of the shaping circuit (shaper) was 1 ms. A 1 MΩ resistor (*R_f_*) and a 10 pF capacitor (*C_f_*) were placed in the feedback loop of the CSP circuit (Figure 4). The output pulses were recorded on an oscilloscope and saved as CSV files for postprocessing in a program written in python. The sampling frequency of the signal was 10 MS/s.

The first detector stage generates a stochastic sequence of current pulses whose amplitude depends on the intensity of the measured radiation (1) [38]
(1)$$V_{o}=\frac{Q_{i}}{C_{f}}=-\frac{1}{C_{f}}\int_{0}^{t}I_{ph}dt$$
where:−1/*C_f_*—charge sensitivity;*Q_i_*—charge forcing;*t*—integration time.

The resistor *R_f_* allows the capacitor *C_f_* to discharge and sets the upper frequency of the circuit.

The collected data were saved in a CSV file and imported for analysis. The comparator trigger threshold and Savitzky–Golay filter (SG filter) parameters were defined in the software (window width and polynomial degree). 

After processing the pulses, the program returned the pulse area bounded by the comparator threshold, the area of the square pulse described on the input signal, the amplitude of the signal, and the width of the square pulse described onto the original signal. All operations were performed on the signal after filtering with the SG filter [42,43] with a window width of 199 samples and a first-order polynomial. The trigger threshold value for the comparator was 0.65 V, and the hysteresis was 0.35 V. For each processed pulse, a graphic was generated to verify the correctness of the signal processing algorithm, as shown in Figure 5.

During forced air measurements, 100 detector frames were collected in CSV files. The results were compared to 161 pulses with particle size analysis. Each frame consisted of 14,000 samples, which were recorded on an oscilloscope. The ‘AS one’ sensor was tested without forced air flow while located in the geometric center of the chamber. The powder was dispensed using a pneumatic cannon.

## 3. Results and Discussion

### 3.1. Characteristics of the Measurement Powder

Particle size analysis of the MS1 steel was carried out on a 100 g sample. The powder had not been previously treated by the SLS/SLM process. The results of the sieve analysis are shown in Figure 6. The distribution shows that the highest percentage of powder was below 45 µm.

Microscopic images of the MS1 powder were taken as shown in Figure 7. The images show that most of the powder was spheroidal in shape [16]. Grains with shapes other than spheroidal were also present in the powder volume [15]. In Figure 7a, small spheroidal shaped grains on the surface of the powder, called satellites, are observed.

On the MS1 powder particle at high magnification (Figure 7b), irregularities resulting from dendritic segregation formed during the powder atomisation process are observed [15].

Based on the Navier–Stokes equation, the particle drop velocities were calculated according to Formula (2)
(2)$$v_{st}=\frac{\left(\rho_{s}-\rho_{c}\right)*g*D^{2}}{18*\eta }$$
where:*g*—acceleration;*ρ_s_*—density of the material;*ρ_c_*—density of the liquid;*η*—dynamic Newtonian viscosity (of air).

For the four materials most commonly used in the SLS/SLM process. The falling velocity as a function of particle size is shown in Figure 8.

Table 1 and Table 2 show the falling velocity values for the four materials and the particle sizes of 25 µm (Table 1) and 45 µm (Table 2).

### 3.2. Characteristics of the SDS198 Modules

From the measurements made with the SDS198 modules, the time after which the mass concentration of MS1 steel returned to its minimum value was determined. The sensors were operating in the continuous mode. The measured values are shown in Figure 9.

All samples collected are shown in Figure 10. Each point in the figure is the average concentration recorded by a given sensor over 1 h.

The measurement was repeated by dosing powder into a test chamber with forced air circulation. The results are shown in Figure 11. 

After the measurements, the particle fall time resulting from the decrease in dust concentration and the Navier–Stokes equation was determined. The calculated fall time from the Navier–Stokes equation was several minutes, and the determined time was over 8 h. The dust concentration dropped to zero despite the additional forced air flow applied. This means that the fans sucking air in the SDS198 modules were not adapted to measure particles with high density. The dust sensor measurements varied due to the uneven distribution of dust in the test chamber. Increased S2 sensor readings show that the position of the air intake was important for particle suction. The producer declares that the module can measure particles from 1 to 100 µm, but in the sensor characteristics chart, neither the shape of the particles nor their density, which can be sucked to the detector chamber [5], was described.

After preliminary analysis, the detector and measuring chamber were re-designed by selecting a fan with a flow of 9.85 m^3^/h, which allowed sucking in particles of much higher density. The applied signal detection solution is a combination of synchronous detection [44,45,46] used in optical signal detection systems [38] and comparator, which is commonly used in multichannel counters of high-energy particles, e.g., in Geiger–Muller counters [47]. In contrast to high energy particle counters and multichannel analysers, it is not necessary to determine the upper operation threshold of the comparator because the system is optically isolated from the environment.

### 3.3. Analysis of the Output Pulses

In order to filter the signal, an FFT analysis was carried out to determine if there is noise in the output signal in the pass band of the system. FFT analysis of Figure 12b shows that at the bandwidth of the measuring system of 16 Hz, the following harmonic frequencies appeared 63 kHz, 126 kHz and 190 kHz.

The recorded signal with no frequencies above 16 Hz is shown in Figure 13a. The second filtration algorithm used was the Savitzky–Golay filtration algorithm. As shown in Figure 12b, in order for the amplitude of the signal without frequencies above 16 kHz to be equal to the amplitude of the signal after filtration with the Savitzky–Golay algorithm, a window of 596 samples should be assumed with a data record length of 14,000 samples. 

As can be seen in Figure 13a,b, the Savitzky–Golay algorithm improved the signal-to-noise ratio much better. The Savitzky–Golay filtering algorithm reduces the noise and allows an increase in the signal-to-noise ratio, which is very beneficial when detecting smaller dust particles.

In order to be able to implement the hardware filtering of the signal in the form of a filter, it was decided to adopt a window of 199 samples (Figure 13b) for the filtering algorithm and a first-degree polynomial. The comparison of the input signal area of the limits of the comparator action and the rectangular pulses showed that they are not related to each other, and the ratios of the values have a normal distribution in the tested sample, which is presented in Figure 14.

After preliminary analysis of the pulses, the average amplitude of the obtained pulses was determined. The detection threshold for the comparator was set to 1/3 of the average amplitude value based on literature data [38]. The collected data were tabulated in a histogram containing six columns in order to compare the obtained results collected from the detector output (Figure 15) to the data obtained from the sieve analysis. 

On the basis of the sieve analysis and the amplitude of the pulses obtained from the detector, the relationship between the particle size and the amplitude of the output pulses from the detector *d*(*V_o_*) was determined, as shown in Figure 16. The relation between the amplitude of output pulses from the detector and the particle size of the powder is exponential [40].

After determining the function *d*(*V_o_*), the equation describing the mass concentration was determined on the basis of the diameter of the grains referred to a sphere. This reference was possible because the atomisation process ensures the spheroidal shape of the powder particles. The equation representing the mass concentration to the number of particles and their diameter is presented in Equation (3)
(3)$$p_{n}=\overset{n}{\underset{k=0}{\sum }}\left(\frac{\frac{4}{3}\pi *{\left(\frac{d\left(V_{o}\right)}{2}\right)}^{3}*\rho *10^{-9}}{V}\right)$$
where:*p_n_* [µg/m^3^]—mass concentration;*d*(*V_o_*)—correlation between pulse amplitude and particle diameter;ρ [kg/m^3^]—density of the material;*V* [m^3^]—volume of aspirated air.

The tests were conducted in order to determine the requirements for further hardware implementation of the solution. The analogue implementation forces the use of input signal filtering to realize synchronous impulse detection. The solution using a comparator for impulse detection requires analogue filtering of the signal to eliminate the influence of noise and to narrow the comparator hysteresis. 

The linear response of the system is possible in the photoconductive mode of the detector [38,48]. 

The linear response of the system based on extrapolation is shown in Figure 17.

The error analysis, i.e., the difference between the percentage of individual fractions from the sieve analysis and the percentage distribution of pulses in given intervals, is shown in Figure 18. It does not exceed 15% over the entire measurement range.

As shown in Figure 18, for larger particles there was a paucity of pulses in the 45 µm range. For particles smaller than 45 µm, the measurements showed an excess of pulses coming from smaller particles. Taking into consideration the accuracy of the sieve analysis and errors resulting from coincidence errors [49], it can be concluded that the inaccuracies are due to the difficulty of aspirating large powder particles with considerable density [18]. In addition, inaccuracies of the measuring system are also due to the high triggering threshold of the comparator, as can be seen in Figure 19a,b where two pulses are treated as one with higher amplitude.

## 4. Conclusions

The results of sieve analysis and automatic measuring methods are not identical to weighing methods [33]. The derived and measured particle sizes may differ depending on the method of analysis. For the sieve analysis, it is related to the mesh size of the sieve. In the case of optical methods, which do not allow for distinguishing the shape of the powder particles, the diameter is related to the diameter of a sphere, so it is assumed that the particles are perfect spheres, from which errors in measurement may arise. In the case of larger powder particles, the aspect of shape can be neglected [50]. 

The settling time calculated from the Navier–Stokes equation was only suitable for the rough estimation of the powder settling velocity due to the assumed conditions. Unfortunately, the simplified equation does not take into account interparticle interaction and assumes laminar flow, which is difficult to obtain in real conditions. The SDS198 modules were not suitable for measurements of particles with high densities, as the air suction velocity was insufficient, as demonstrated above.

Savitzky–Golay filtering significantly improved the signal-to-noise (S/N) ratio, which was very beneficial. The elements used and the noise level allowed for the detection of smaller sizes (15 µm). The size of detected particles depended on the slope of the characteristic *d*(*V_o_*) and on the value of noise in the system. The application of the photoconductive mode in the detector would have enabled the detection of particles smaller than 15 µm and obtained a linear response for the detector.

Measurements have shown that the ‘AS one’ detector is effective for measuring high density powder particles in the 15–90 µm range. The detector has a small size and can be used in applications where the dust concentration is measured continuously. 

SG filtering significantly improved the S/N ratio, which was very useful for the detection of smaller powder particles.

In a practical application, it would be advantageous to use a fourth or higher order Bessel filter. The comparator signal could be used for synchronous pulse detection or implementation of a digital peak detector [51]. The time length of the comparator pulse could be used to implement a coincidence error correction algorithm. The proposed simplified detection methods for synchronous [52] and phase sensitive [53] is sufficient for the detection of particles above 15 µm.

From the results obtained, it can be seen that it is necessary to work on the dynamics of the particle intake or the shape of the duct to optimise the air flow, or the shape of the chamber to avoid coincidence errors. It is crucial to select the elements of the optical path, filtration ensuring good S/N ratio and the threshold of activation of the comparator. This procedure can be used to calibrate dust sensors with powder particles of a known grain size distribution.

The use of commercially available electronic components in the sensor design reduced the sensor production costs. The operation of the sensor in particle counter mode and the ability to analyse single powder particles places the device between low-cost dust sensors and professional particle analysers. Because of the choice of the laser diode and the lack of additional optical elements, the cost of manufacturing a sensor based on the presented solution is in the range of USD 500–1000, offering the functionality of devices whose purchase cost is over USD 20,000.

## Figures and Tables

**Figure 1 sensors-22-07614-f001:**
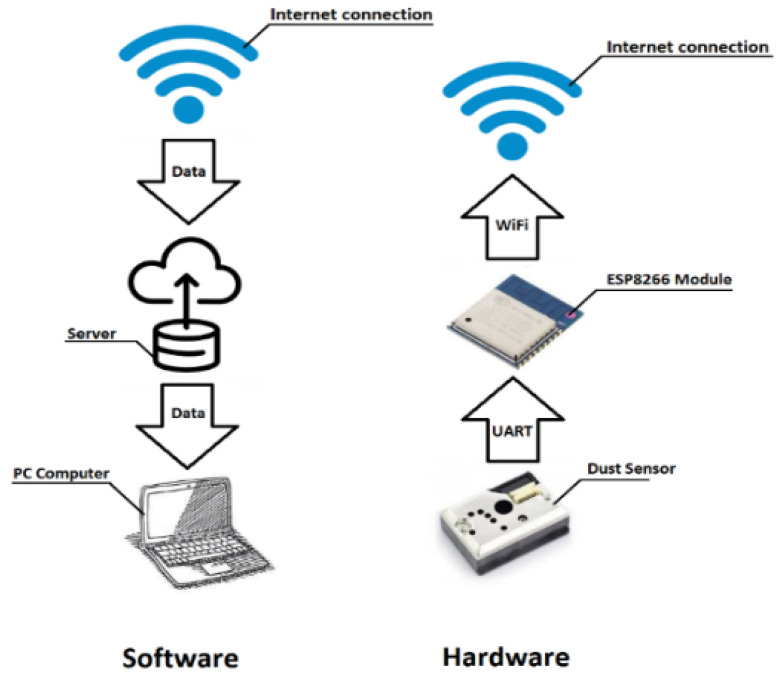
Operation diagram of measuring stand.

**Figure 2 sensors-22-07614-f002:**
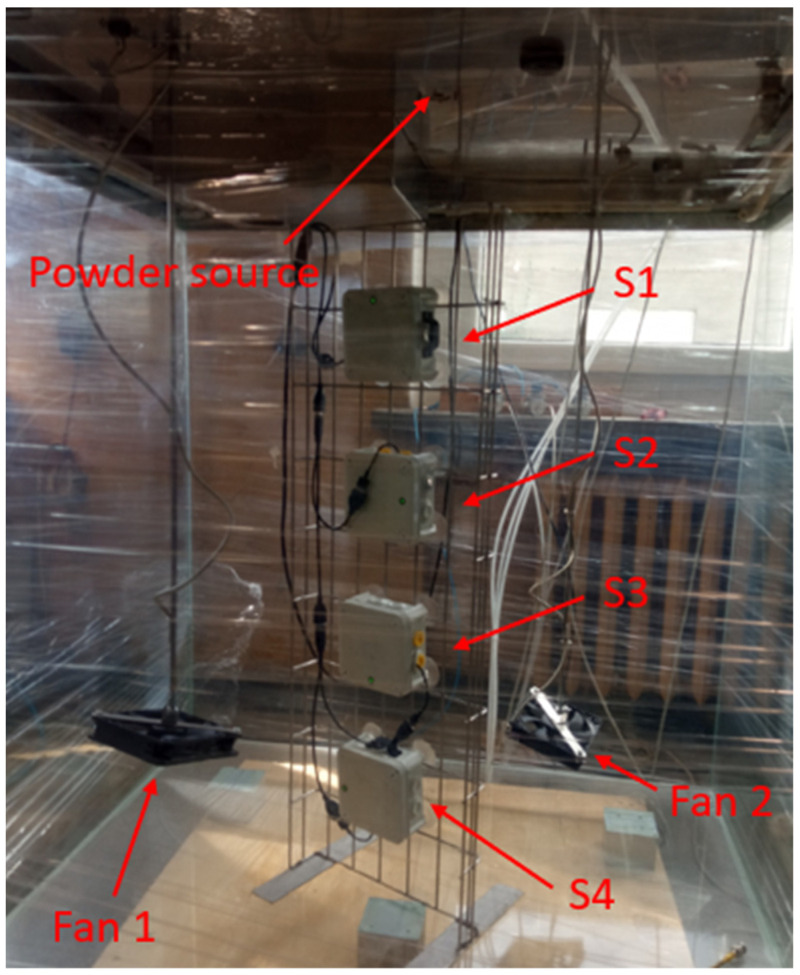
Measuring stand for dust measurements.

**Figure 3 sensors-22-07614-f003:**
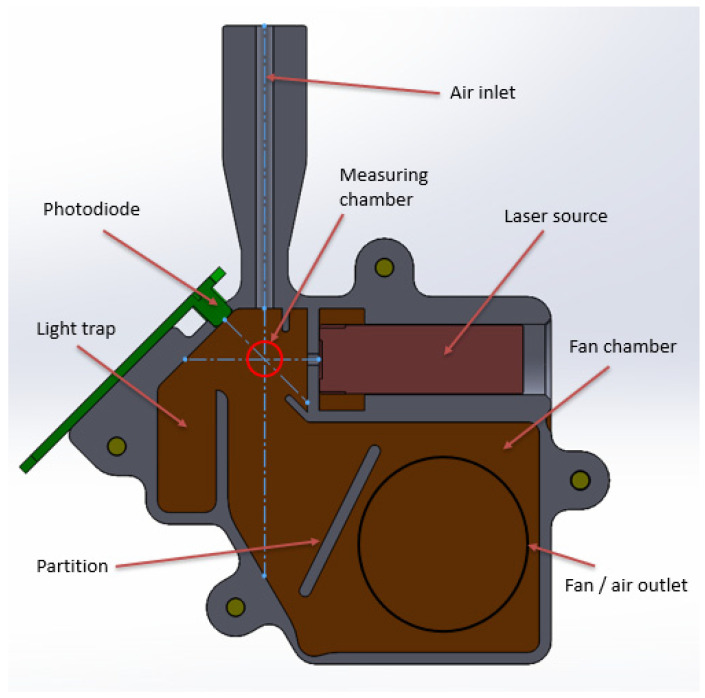
Cross-section of the dust sensor.

**Figure 4 sensors-22-07614-f004:**
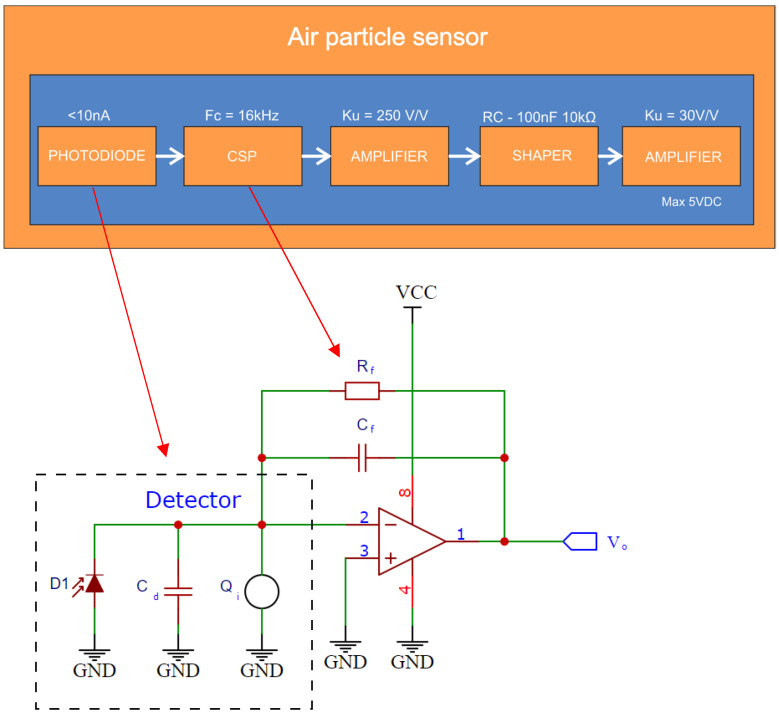
Operation diagram of the dust sensor.

**Figure 5 sensors-22-07614-f005:**
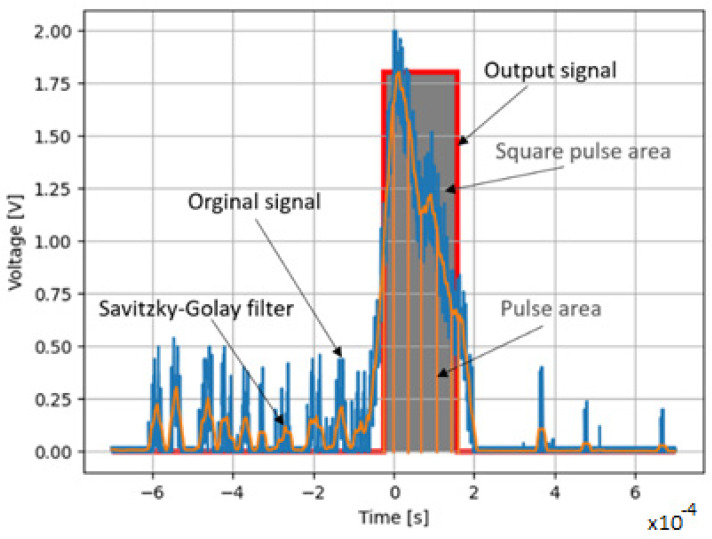
Processed pulse after analysis.

**Figure 6 sensors-22-07614-f006:**
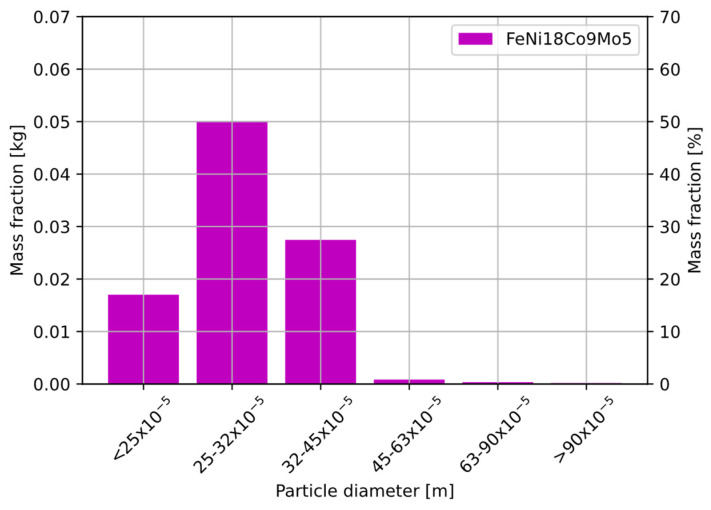
Particle size analysis for FeNi18Co9Mo5 (maraging MS1).

**Figure 7 sensors-22-07614-f007:**
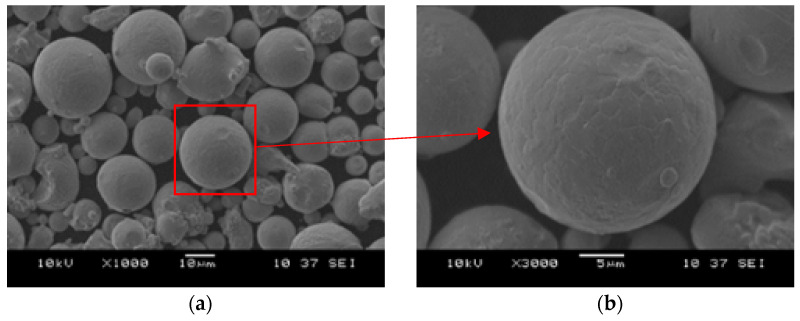
SEM microscopy images (**a**) maraging MS1 powder (**b**) maraging MS1 particle.

**Figure 8 sensors-22-07614-f008:**
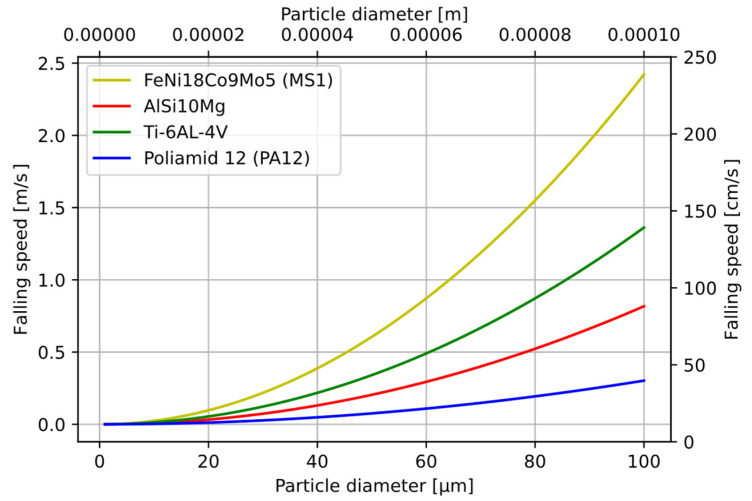
Particle fall velocity calculated from Navier–Stokes equation.

**Figure 9 sensors-22-07614-f009:**
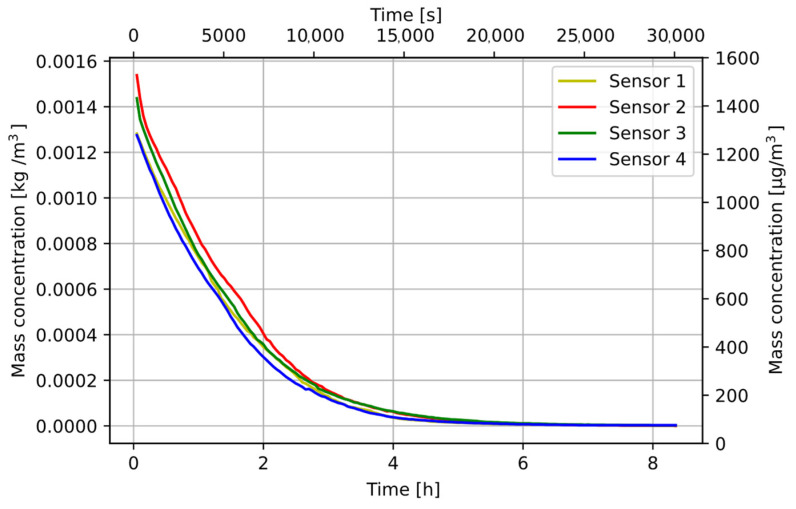
Change in MS1 steel concentration in the air (all samples).

**Figure 10 sensors-22-07614-f010:**
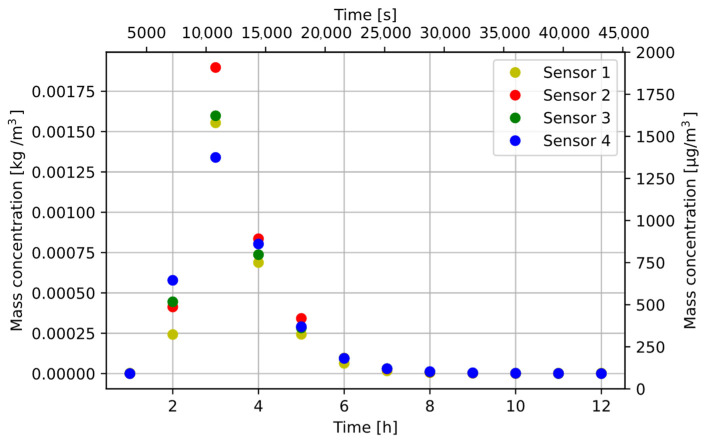
Change of MS1 steel concentration in the air—without forced air flow in the chamber.

**Figure 11 sensors-22-07614-f011:**
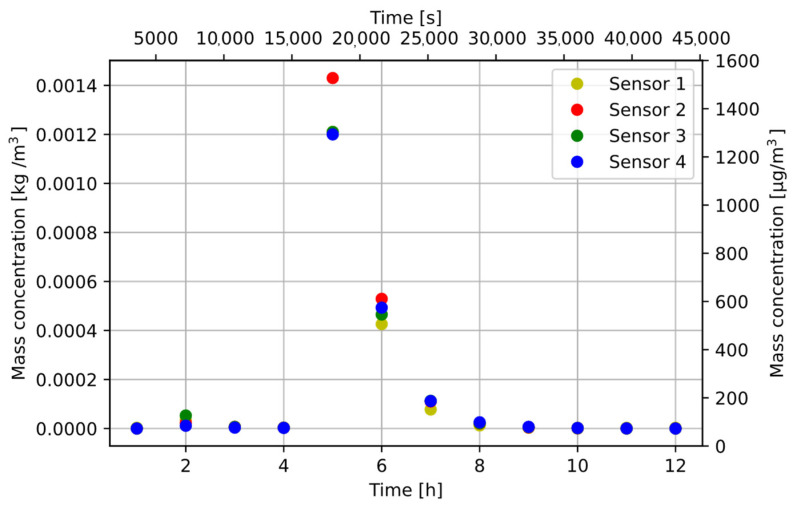
Change of MS1 steel concentration in the air—with forced air flow in the chamber.

**Figure 12 sensors-22-07614-f012:**
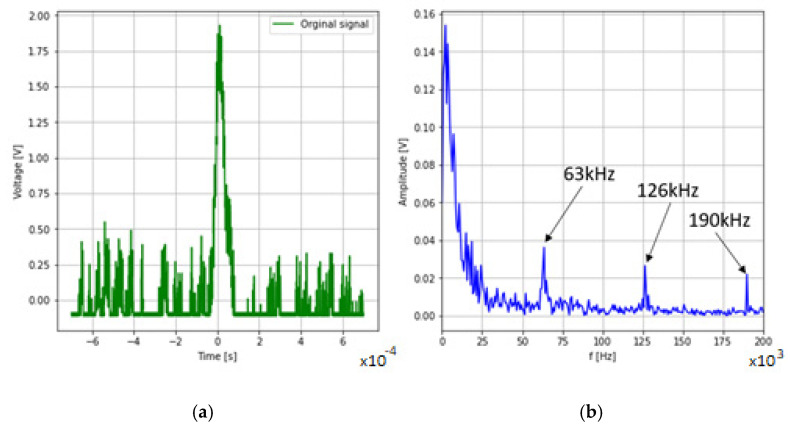
Raw detector pulse (**a**) and frequency domain analysis (**b**).

**Figure 13 sensors-22-07614-f013:**
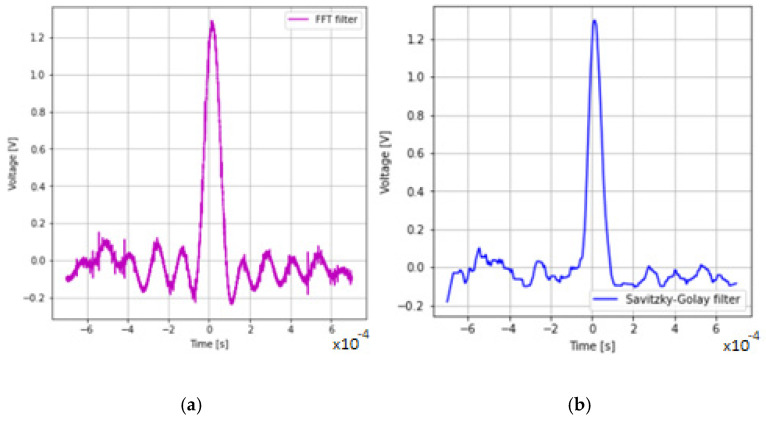
Comparison of filtration after FFT filtering (**a**) and SG filtering (**b**).

**Figure 14 sensors-22-07614-f014:**
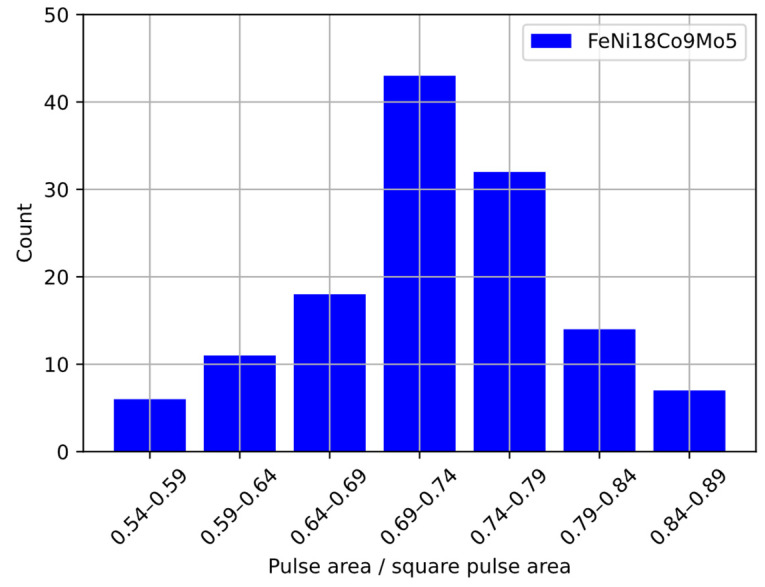
Ratio between the area of the comparator-limited pulses and the square pulses.

**Figure 15 sensors-22-07614-f015:**
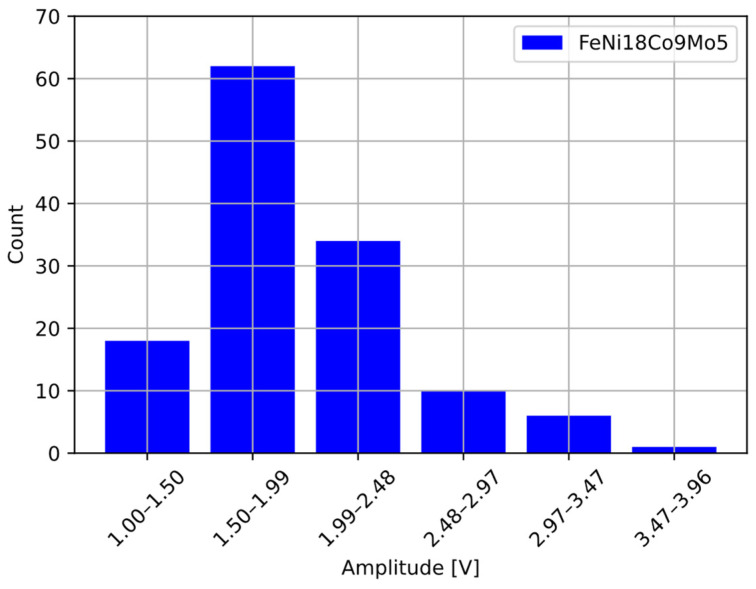
Analysis of the collected pulses from the detector.

**Figure 16 sensors-22-07614-f016:**
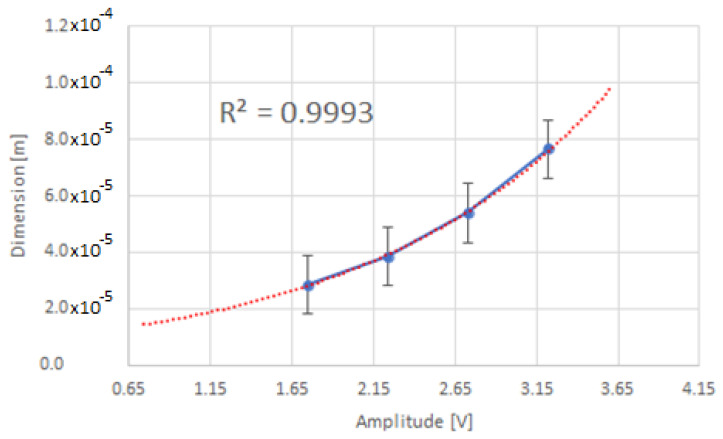
The relationship between pulse amplitude and particle size.

**Figure 17 sensors-22-07614-f017:**
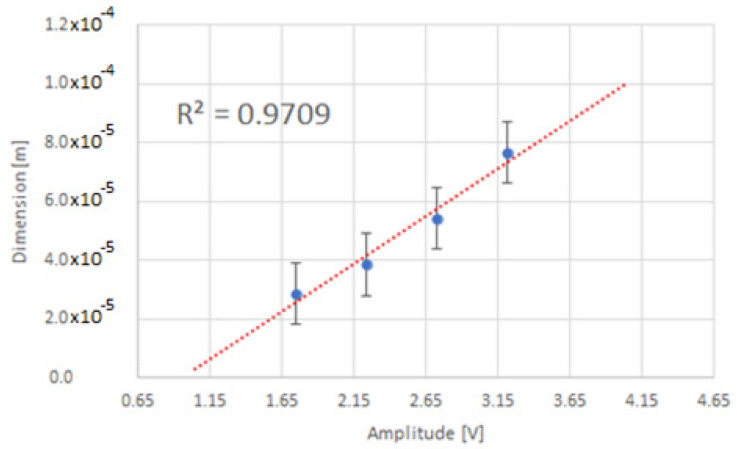
The relationship between pulse amplitude and particle size (linear extrapolation).

**Figure 18 sensors-22-07614-f018:**
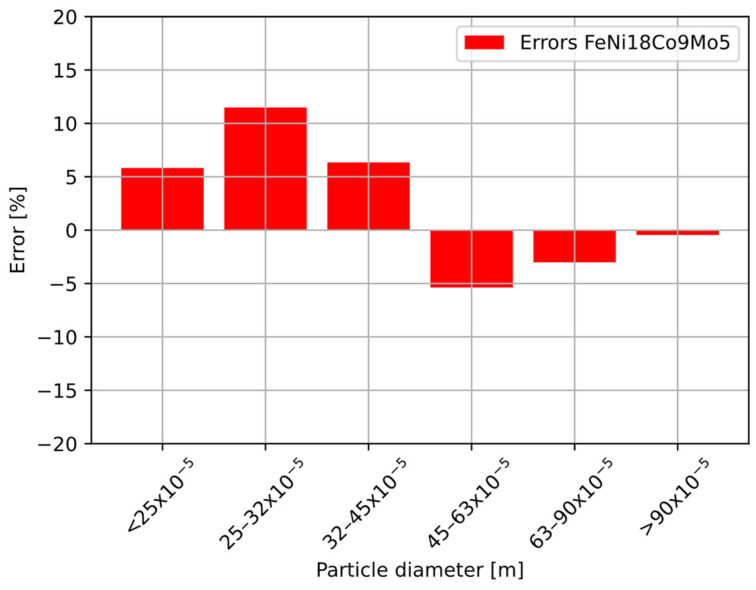
Error distribution for different particle sizes.

**Figure 19 sensors-22-07614-f019:**
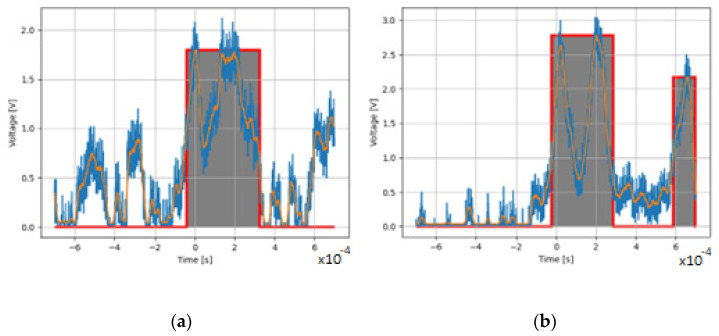
Comparator operation errors for two similar series pulses.

**Table 1 sensors-22-07614-t001:** Particle fall velocity for 25 µm diameter particles calculated from Navier–Stokes equation.

Material	Dimension [µm]	Falling Speed [m/s]
Polyamide 12 (PA12)	25	0.018
AlSi10Mg	25	0.051
Ti-6AL-4V	25	0.085
FeNi18Co9Mo5	25	0.151

**Table 2 sensors-22-07614-t002:** Particle fall velocity for 45 µm diameter particles calculated from Navier–Stokes equation.

Material	Dimension [µm]	Falling Speed [m/s]
Polyamide 12 (PA12)	45	0.061
AlSi10Mg	45	0.165
Ti-6AL-4V	45	0.275
FeNi18Co9Mo5	45	0.490

## Data Availability

The data presented in this study are available on request from the corresponding author.

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
