# Peer review of "Dust Particle Counter for Powder Bed Fusion Process"

_sensors, 2022, doi:10.3390/s22197614_

Round 1

Reviewer 1 Report

General comments

Research and development relates to a very important issue, and it is worth publishing. The material is adequate for the scope of the journal. Unfortunately authors need more work before it can be accepted for the publication.

Two main issues- the text needs very careful proof-reading:

- the text needs careful spelling and grammar checks (at least using the embedded MsWord possibilities for English spelling and grammar checks);

- ideally the text should be also proof-read by the person with English as the mother tongue, correcting the style and terms (some of the used terms are not OK).

All in all, the paper has a very good potential, it is devoted to a very important issue and leading to the possibilities for future commercialization. Can be accepted after careful revisions.

Particular comments

(i) Proof reading is absolutely essential.

There are repetitions of the same words separated in the text less than on-two other words (for example p. 4, line 176). There are repetitions of the whole sentences (see page 3, lines 98-101). There are  full stops (‘dots’) missing at the ends of sentence, and there are some in the middle (p.2, lines 91 and 93). Many more cases were found in just first 6 pages.

I did not have any time to proof-read the text, but even ordinary reading show a lot of misprints etc.

(ii) Title/Abstract/text

- Title: ‘3D printing’ is not an accepted term with the specialists. It should be additive Manufacturing.

- Moreover, the material is mainly related to the powder-bed AM methods. At least it should be made clear in the Abstract, but may be it is also better to change the title to ‘Dust particle counter for powder bed additive manufacturing’- up to authors. But if the general title is chosen (‘Dust particle counter for additive manufacturing’ or alike) authors need to add more general consideration in the Introduction, related to other methods.

(iii)  Introduction, 1st phrase ‘…(FDM)…’

Why??? FDM is mainly used for polymer AM 8and with the polymer ‘filaments’ as raw material, not powder)!- see (ii) above

(iii) page 2, last lines of the text

‘…degree of (surface) smoothness…’ – you possibly mean the surface roughness?

(iv) Materials and methods, pp 4 and 5

a) The structuring of the text should be improved.

It is better to start with the general layout of the ‘measurement process’ by moving current Fig. 2 as Fig 1, and describing the functionality of the elements. Otherwise it is unclear what are the ‘SDS 198’ and ‘ESP 8266’ (these are mentioned on page 4, much earlier than the diagram of current Fig 2). And this description should be more careful and more detailed.

b) Photo of the setup (chamber) as it is in present Fig. 1

The description should be much more careful and precise.
page 4, line 183 claims that the sensor is  ‘..in the center of the chamber..’ There is nothing present in the center of the chamber in the photo in Fig 1!page 5, line 188 claims that the fans are in the chamber, but there is nothing visible in the photo!

(v) page 5, line 201: ‘… polycatid (PLA…’- do you mean polyACtid?

(vi) page 7, line 241, the example of the style/definitions/terms issues:

‘…The collected impulses saved in csv file format…’

Impulses cannot be collected in the text format- it is unclear what was meant.

Descriptions of the functioning, hardware, software should be improved.

(vii) 5. Patents

Listing of the patents without references to them in the text leads to questions…

a) Why do you mention them if you are not using them in any way throughout the text?

b) if you refer to the patents with the titles that are not in English, you need to provide at least the translation  of the title in the brackets (…).

c) if the English translations of these patents (or at least of the corresponding Abstracts from the patents) are not openly available, the value of such references in the international journal publication is questionable.

Please check with the scan of the paper with some handwritten remarks (at least up to page 7) showing all I was able to see and comment upon.

Author Response

We would like to thank Reviewers for taking the necessary time and effort to review the manuscript. We sincerely appreciate all your valuable comments and suggestions, which helped us in improving the quality of the manuscript.

(i) Proof reading is absolutely essential- corrected

(ii) Title/Abstract/text - thanks for the suggestion the title has been changed to “Dust particle counter for powder bed fusion process”.

(iii) Introduction, 1st phrase ‘…(FDM)…’ - I would like to emphasize that with the increase in popularity of a particular technology, there is an increase in the number of publications related to the impact of the technology on the health and life of workers

(iii) page 2, last lines of the text -  roughness - corrected

(iv) Materials and methods, pp 4 and 5

a) corrected  - Changed the sequence of fig. 1 and fig. 2, added description of esp8266 module (NodeMCU v3)

b) corrected - photo has been changed

(v) Unfortunately no, the correct form in my opinion is “polylactide”.

(vi) corrected  - “The collected data saved in csv file format”

(vii) 5. Patents

a) corrected

b) corrected

c) references removed from text 

Please check with the scan of the paper with some handwritten remarks (at least up to page 7) showing all I was able to see and comment upon. - unfortunately I did not receive the attachment

Reviewer 2 Report

The paper presents a dust detector based on laser system used in applications of dust measurement in powder bed processes. In my opinion, the work is of particular interest for the development of a dedicated sensors for 3D printing process. The study is well-described and experimental results are nicely demonstrated. The current paper is worthy of investigation and organization. The following comments need to be addressed to be accepted for publication.

1.It is pointed out that  'AS one' is a low-cost detector. It needs to be compared with other detectors from the cost perspective.

2. The detector is designed for  3D printing process, It is recommended to carry out the test under the condition of 3D printing.

3. The principle of the dust detection sensor needs to be explained.

4. In line 311, "SDS192 modules" was not mentioned above.

5. In Figure 9 and Figure 10, the authors should explain the changes of mass concentration of  the detectors(S1-S4). In figure9 the value of S4 is higher than that of other parameters. But in Figure 10, the value of S2 is higher then the others between 3-5 hours.

6. I am confused that different particle sizes are mixed together, how to accurately obtain the number of particles with different diameters?

7. The text should be proofread and corrected. 

  Examples: Line 228: f should be the subscript.

                  Line 279, 280: s, c should be the subscript.

                 Figure 14.  The label on the left is different from the label on the right.

                 Equation (2). The units should not appear in the formula.

Author Response

We would like to thank Reviewers for taking the necessary time and effort to review the manuscript. We sincerely appreciate all your valuable comments and suggestions, which helped us in improving the quality of the manuscript

1. added in chapter conclusions

2. presented article is part of the research on the final solution, research under normal operating conditions will continue in further research

3. The principle of the dust detection are described in the chapter 1.2 “Dust detection methods”.

4. I meant SDS198 - corrected

5. corrected - labels were swapped during data development

5a. added under fig. 11 - “Dust sensor measurements varied due to the uneven distribution of dust in the test chamber”.  

6. The sensor was scaled for the powders used in the real manufacturing process (Selective Laser Sintering / Melting processes). Ideally scaling and choice of correction coefficients shall be done for various particle sizes individually, but finally it would be necessary to scale it for the powder used in practice. The other way would be to set coefficient relevant for the fraction that has biggest share in the powder (in case of used metallic powders for SLS/SLM it would be 45-55um, which is more than 50% in the volume of the powder used). This could also lead to omitting important information on the presence of other fractions. As the developed and described sensor will be used in practice, for monitoring of SLS/SLM laboratories / rooms, it was scaled with described in the paper utilities for the powders used in practice. The results of the sensor readouts were compared with the powder characteristics charts supplied by powder suppliers and sieve analysis of the used powders.

The amount of each fraction was determined by sieve analysis

It has been proven in the literature that calibration with powder of different granularity can give satisfactory results, e.g. In the publication of

  1. Pedersini, „Improving a Commodity Dust Sensor to Enable Particle Size Analysis”, IEEE Trans. Instrum. Meas., t. 68, nr 1, ss. 177–188, 2019, doi: 10.1109/TIM.2018.2834178.

7.  corrected, 

Figure 14.  The label on the left is different from the label on the right.

– the second axis is in percentages therefore the values do not correspond to the left axis – axis removed from fig. 14 and fig. 15

Round 2

Reviewer 2 Report

I am recommending this manuscript for publication. Well done!